# A link between smoking behaviors and the risk of hidradenitis suppurativa in diabetic patients

**Hwa Jung Yook**[1], **Esther Kim**[1], **Yeong Ho Kim**[1], **Gyu-Na Lee**[2], **Kyungdo Han**[3], **Ji Hyun Lee**[1]*

**1** Department of Dermatology, Seoul St. Mary's Hospital, College of Medicine, The Catholic University of Korea, Seoul, Republic of Korea, **2** Department of Biomedicine & Health Science, The Catholic University of Korea, Seoul, Republic of Korea, **3** Department of Statistics and Actuarial Science, Soongsil University, Seoul, Republic of Korea

* yiji1@hanmail.net

## Abstract

### Introduction

Hidradenitis suppurativa (HS) is a chronic, inflammatory skin condition characterized by painful, recurrent abscesses and tunnels under the skin. It is influenced by a complex interplay of genetic, environmental, and metabolic factors. Smoking, diabetes mellitus, obesity, and other metabolic disorders have been identified as risk factors for HS, potentially exacerbating the severity and progression of the condition. Given the higher prevalence of HS in individuals with type 2 diabetes mellitus (T2DM), understanding modifiable risk factors such as smoking is crucial for improving patient outcomes.

### Objectives

To investigate the association between changes in smoking intensity and the risk of HS in patients with T2DM, with the goal of elucidating how smoking contributes to the development or worsening of HS in this high-risk population.

### Methods

This retrospective cohort study analyzed data from the Korean National Health Insurance Service, comprising 1,705,427 participants. The study examined smoking status, changes in smoking intensity, and the incidence of HS in individuals with T2DM, adjusting for potential confounders such as age, gender, and comorbid conditions.

### Results

The study found a 23.6% increased risk of HS in individuals who continued smoking compared to nonsmokers (adjusted hazard ratio (aHR), 1.236; 95% confidence interval (CI), 1.075–1.421). Additionally, participants with increased cigarette consumption

**Data availability statement:** The data in this study was obtained from KNHIS where licence may apply. Further enquiries can be directed to the corresponding author. We have identified and included a non-author institutional point of contact to ensure long-term data availability. The details are as follows: Institutional Point of Contact: Research Office; Contact Information: dermaseoul@gmail.com

**Funding:** The author(s) received no specific funding for this work.

**Competing interests:** All authors do not have any conflict of interests.

had a 28.5% higher risk of HS compared to nonsmokers (aHR, 1.285; 95% CI, 1.048–1.577). However, no significant differences were observed in the association between changes in smoking intensity and the risk of HS when stratified by BMI.

## Conclusions

This study highlights the association between smoking and the increased risk of HS in individuals with T2DM, underscoring the importance of smoking cessation as a potential strategy for mitigating HS risk in at-risk DM populations. Additional research is needed to further explore the mechanisms by which smoking exacerbates HS in T2DM and to identify effective interventions for this group.

## Introduction

Hidradenitis suppurativa (HS), also known as acne inversa, is a persistent inflammatory skin condition characterized by recurrent painful lesions [1,2]. These lesions are deep-seated and inflamed, primarily occurring in regions of the body with apocrine glands, such as the axillae, inguinal, and anogenital areas. The annual incidence of HS falls between 4 to 10 cases per 100,000 people during the period from 1968 and 2008 [3], with a prevalence ranging from 0.2% to 4% [1,4]. The typical age of onset is approximately 23 years old [5]. The condition affects females more frequently, with a male-to-female ratio ranging from 1:2.7 to 1:3.3 in European and North American populations [5,6]. Conversely, Asian populations exhibit a reverse pattern, with a female-to-male ratio of approximately 1:2 to 1:5.6 [7–11].

While the exact cause of HS remains unknown, it is recognized as a multifactorial condition influenced by genetic, metabolic, and environmental factors. HS is more commonly observed in females and often improves during pregnancy, with premenstrual flares and early onset before menopause suggesting a potential hormonal or metabolic origin [12]. Additionally, HS has been associated with smoking, increased body mass index (BMI), metabolic syndrome, inflammatory bowel disease, and polycystic ovary syndrome [13].

Type 2 diabetes mellitus (T2DM) is a chronic disease that develops when the body cannot effectively produce and use insulin [14]. Several studies have reported a notable association between HS and DM [15–18]. However, the precise occurrence of DM in HS patients remains uncertain, with reported rates ranging from approximately 4% to 33% [12,19,20]. Two meta-analyses found a 1.69–3.00 fold increase in the odds of DM in HS populations [12,21,22]. Despite these findings, a causal relationship between DM and HS has yet to be established [12,23]. It has been hypothesized that HS might contribute to DM through chronic systemic inflammation leading to elevated levels of tumor necrosis factor-alpha (TNF-α). Conversely, insulin resistance in DM may predispose individuals to HS [23].

Smoking, in particular, has been strongly linked to HS, with a large US-based retrospective cohort study reporting a twofold increase in HS incidence among smokers [24], and a meta-analysis indicating that HS patients are four times more

likely to be smokers than the general population [25]. Smoking rates of 70–90% have been observed in HS populations [26,27], and surveys of disease severity have shown a positive correlation between smoking and the severity of HS [28]. However, it remains unclear whether smoking is merely a risk factor or directly contributes to the pathogenesis of HS [29]. Smoking cessation appears to reduce the risk of adverse outcomes in HS patients [30], but limited research has explored how changes in smoking intensity influence the risk of HS, particularly in individuals with comorbidities such as DM. DM and smoking are known risk factors for HS, but their combined impact on HS development or progression remains poorly understood. This study aims to explore how changes in smoking behavior influence HS risk in DM patients, using a nationally representative Korean claims database, to identify modifiable risk factors in this high-risk population.

## Materials and methods

### Study design and population

This retrospective cohort study utilized population-level data from the Korean National Health Insurance Service (KNHIS), a compulsory universal insurance system providing coverage for all individuals in Korea. The standardized protocol for data acquisition by the KNHIS has been previously described [31].

The KNHIS database includes demographic information linked to healthcare claims, data from the national health checkup program, and self-reported, health-related questionnaires. It also contains measurements of height, weight, and blood pressure, as well as results from laboratory tests, including blood and urine analyses.

This study included participants with complete smoking status data from their initial health checkup conducted between 2009 and 2012, along with a second health checkup within a two-year interval. Specifically, the presence of diabetes mellitus was determined for all participants. The cohort consisted of patients who had been diagnosed previously with T2DM and those who were newly diagnosed during health checkups. Initially, 2,746,078 patients with T2DM were included. Subjects were excluded from the study if their records indicated any of the following: 1) the second health checkup not been conducted (n = 915,219), 2) under the age of 20 (n = 4), 3) a diagnosis of HS before the first health checkup (n = 2,280), 4) the development of incident HS or death during the one-year lag period (n = 11,246), or 5) having missing data for at least one variable (n = 111,902). In total, 1,705,427 participants were included in the main analysis (Fig 1). The baseline for the study was established at the date of the second health checkup, and eligible participants were followed until the date of death or the date of HS diagnosis or December 31, 2018.

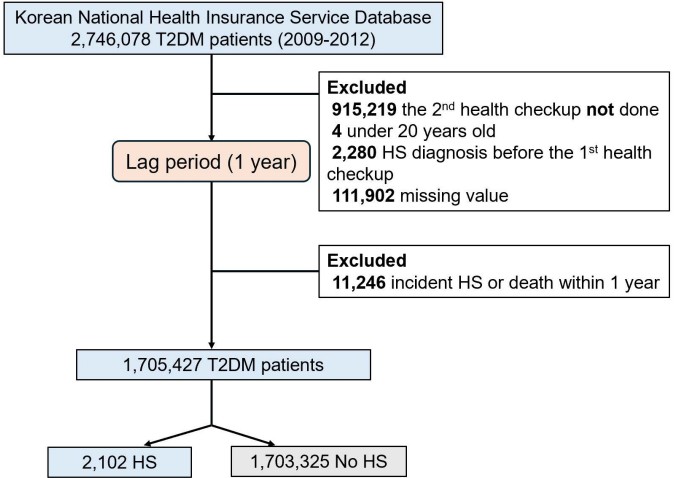

**Fig 1. A flow diagram of study population selection.**

This study received approval from the Institutional Review Board of Catholic Medical Center (IRB No. KC23ZASI0891) and the requirement for informed consent was waived since the study relied on anonymized retrospective data. We accessed the data for research purposes on December 15, 2023.

## Definition of disease

DM was identified based on previously established criteria [32–35]: (1) at least one annual claim for the prescription of anti-diabetic drugs categorized under ICD-10 codes E11–14; or alternatively, (2) fasting plasma glucose level equal to or greater than 126 mg/dL during health examinations. Anti-diabetic medications included insulin, metformin, sulfonylureas, meglitinides, dipeptidyl peptidase 4 (DPP-4) inhibitors, thiazolidinediones, and α-glucosidase inhibitors. HS was identified using the diagnostic ICD-10 code of L73.2. The incidence rate of HS was calculated per 100,000 person-years.

## Definition of a change in smoking intensity

Smoking status was determined through self-reported surveys conducted during the initial health screening assessments and again after 2 years (subsequent health assessment). Data on smoking status and changes in smoking intensity were collected. Participants who answered that they had smoked at least 100 cigarettes in their lifetime were asked about their current smoking status according to the World Health Organization's definition [36]. Among the current smokers, additional information was gathered regarding the duration of smoking and the average number of cigarettes smoked per day. Based on the cigarette smoking status reported during the initial health examination in 2009–2012, participants were grouped into never smokers, former smokers, and current smokers. In this study, changes in cigarette smoking intensity were assessed as the relative change in daily cigarette consumption. We classified participants into seven categories based on the relative change in smoking amount between the initial (2009–2012) and follow-up (after 2 years) health examinations. Participants were grouped into seven categories, defined based on prior research [37–39]. Quitters were individuals who completely stopped smoking, meaning they were smokers at the initial checkup but had quit by the follow up checkup. 50% Reducers were participants who reduced their daily cigarette consumption by 50% or more compared with their initial level. 20% Reducers were individuals who decreased their daily cigarette consumption by 20% to 50% compared with their previous level. Sustainers were those who maintained their daily cigarette consumption or either increased or decreased consumption by less than 20%. Increasers were participants who increased their daily cigarette consumption by 20% or more compared with their previous amount. Never smokers were participants who responded as never smoking at both biennial health examinations. Starters were those who reported being nonsmokers at the first examination but identified as current smokers at the second examination (Fig 2).

## Demographic factors

We collected demographic and health-related information from all participants, including age, smoking habits, alcohol consumption, regular physical activity, height, weight, blood pressure, total serum cholesterol, glucose levels, and medical history related to diabetes. Blood samples for serum glucose and lipid level measurements were obtained after an overnight fast, following quality control procedures in accordance with the Korean Association of External Quality Assessment guidelines [40]. Weight measurements were conducted while participants wore light clothing. Height was measured without shoes, with participants maintaining proper posture (heels, buttocks, shoulders, and head aligned), and recorded in meters to two decimal places. BMI was calculated using the formula: weight (kg)/ height$^2$ (m$^2$) and classified based on the World Health Organization's Asia-Pacific criteria, with a BMI of 25 kg/m$^2$ or greater categorized as obesity [41,42]. Systolic and diastolic blood pressure were measured in a seated position after a minimum of 5 minutes of rest. Regular physical activity was defined as engaging in ≥20 minutes of vigorous-intensity physical activity on ≥3 days per week or ≥30 minutes of moderate-intensity physical activity on ≥5 days per week [43,44]. Alcohol consumption data, including the

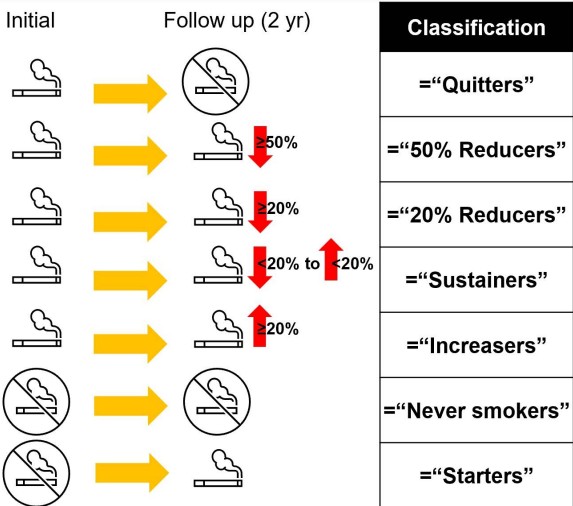

**Fig 2. The classification based on changes in smoking intensity at initial and follow-up assessments.**

frequency of drinking per week and the amount consumed per occasion, were converted into daily alcohol intake using previously established methods [45]. Levels of alcohol consumption were categorized as nondrinking, mild drinking (<30 g per day), and heavy drinking (≥30 g per day).

## Statistical analysis

All statistical analyses were performed using SAS software (version 9.4, SAS Institute, Cary, NC). The cumulative incidence of HS between groups was compared using the Kaplan-Meier method. Adjusted hazard ratios (HRs) and 95% confidence intervals (CIs) for HS risk were estimated using multivariable Cox proportional-hazards models.

Covariates included age, sex, BMI, income, alcohol consumption (categorized as non-drinking, mild drinking, or heavy drinking), regular exercise (yes or no), medical history of hypertension, dyslipidemia, chronic kidney disease (yes or no), fasting glucose levels, duration of diabetes, and the use of oral hypoglycemic agents or insulin.

Additionally, HRs for incident HS associated with changes in smoking behavior were further adjusted for BMI status, categorized as either <25 kg/m$^2$ or ≥25 kg/m$^2$. All statistical tests were two-tailed, and a p-value <0.05 was considered statistically significant.

## Results

### Baseline characteristics of the study population

The cohort was composed of 1,705,427 participants (581,618 males [34.1%], 1,123,809 females [65.9%]; mean [SD] age, 59.03 [11.76] years). Table 1 shows the baseline characteristics during the 2009–2012 health examination according to relative changes in smoking intensity (never smokers, starters, quitters, 50% reducers, 20% reducers, sustainers, and increasers). At the initial health examination, 56.2% of participants were nonsmokers, 19.3% were former smokers, and 24.6% were current smokers. Among smokers, 9.8% were mild smokers, 34.1% were moderate smokers, and 56.1% were heavy smokers. At the second health examination, 4.7% of participants had quit smoking, and 5.2% had reduced their smoking (2.0% in the 50% reducer group and 3.2% in the 20% reducer group). Meanwhile, 3.9% of participants increased the number of cigarettes they smoked during the 2-year interval. Proportions of starter and sustainer groups were estimated as 3.3 (%) and 10.8 (%), respectively.

**Table 1. Baseline characteristics of the study population with DM according to changes in smoking intensity in 2009–2012.**

| | Never | Starter | Quitter | 50% Reducer | 20% Reducer | Sustainer | Increaser | P value |
|---|---|---|---|---|---|---|---|---|
| | (n = 1229850) | (n = 56633) | (n = 79829) | (n = 33327) | (n = 54102) | (n = 184565) | (n = 67121) | |
| **Age, years** | 61.17 ± 11.2 | 54.99 ± 11.22 | 55.97 ± 11.44 | 54.93 ± 11.81 | 52.16 ± 11.25 | 52.5 ± 10.97 | 52.36 ± 11.5 | <.0001 |
| **Sex, male** | 581618 (47.29) | 51392 (90.75) | 72594 (90.94) | 31157 (93.49) | 52093 (96.29) | 178308 (96.61) | 63456 (94.54) | <.0001 |
| **Income, Lowest Quartile 1** | 245419 (19.96) | 11009 (19.44) | 15231 (19.08) | 7223 (21.67) | 10057 (18.59) | 34384 (18.63) | 13209 (19.68) | <.0001 |
| **Alcohol consumption** | | | | | | | | <.0001 |
| None | 841638 (68.43) | 18069 (31.91) | 36417 (45.62) | 11099 (33.3) | 14750 (27.26) | 49286 (26.7) | 18706 (27.87) | |
| Mild | 323478 (26.3) | 29721 (52.48) | 34229 (42.88) | 17984 (53.96) | 30008 (55.47) | 98986 (53.63) | 34146 (50.87) | |
| Heavy | 64734 (5.26) | 8843 (15.61) | 9183 (11.5) | 4244 (12.73) | 9344 (17.27) | 36293 (19.66) | 14269 (21.26) | |
| **Regular exercise** | 292717 (23.8) | 12782 (22.57) | 18990 (23.79) | 7374 (22.13) | 11191 (20.69) | 35977 (19.49) | 12779 (19.04) | <.0001 |
| **Comorbidities** | | | | | | | | |
| Hypertension | 753815 (61.29) | 29290 (51.72) | 42745 (53.55) | 17114 (51.35) | 25604 (47.33) | 87726 (47.53) | 31769 (47.33) | <.0001 |
| Dyslipidemia | 587899 (47.8) | 23018 (40.64) | 34843 (43.65) | 13400 (40.21) | 20921 (38.67) | 70869 (38.4) | 25729 (38.33) | <.0001 |
| CKD | 147074 (11.96) | 3901 (6.89) | 6568 (8.23) | 2377 (7.13) | 2884 (5.33) | 9244 (5.01) | 3514 (5.24) | <.0001 |
| **DM Duration, ≥ 5 years** | 509078 (41.39) | 18452 (32.58) | 26601 (33.32) | 10375 (31.13) | 14951 (27.63) | 51484 (27.89) | 18907 (28.17) | <.0001 |
| **OHA, ≥ 3** | 218704 (17.78) | 9424 (16.64) | 14507 (18.17) | 5672 (17.02) | 8150 (15.06) | 27809 (15.07) | 10423 (15.53) | <.0001 |
| **Insulin** | 108611 (8.83) | 4021 (7.1) | 7339 (9.19) | 2559 (7.68) | 3353 (6.2) | 10885 (5.9) | 4286 (6.39) | <.0001 |
| **BMI, kg/m², mean±SD** | 24.96 ± 3.29 | 24.75 ± 3.23 | 24.89 ± 3.21 | 24.6 ± 3.36 | 24.75 ± 3.36 | 24.69 ± 3.3 | 24.75 ± 3.4 | <.0001 |
| **Waist circumference, cm** | 84.63 ± 8.68 | 85.93 ± 8.16 | 86.4 ± 8.13 | 85.89 ± 8.34 | 85.98 ± 8.3 | 85.92 ± 8.21 | 85.97 ± 8.45 | <.0001 |
| **SBP, mmHg** | 128.34 ± 15.3 | 126.42 ± 14.7 | 127.13 ± 14.72 | 126.87 ± 15.03 | 126.67 ± 14.55 | 126.66 ± 14.56 | 126.64 ± 14.8 | <.0001 |
| **DBP, mmHg** | 77.82 ± 9.85 | 78.21 ± 10 | 78.4 ± 9.92 | 78.42 ± 10.02 | 78.84 ± 9.94 | 78.78 ± 9.92 | 78.76 ± 10.03 | <.0001 |
| **Laboratory findings** | | | | | | | | |
| Fasting glucose, mg/dL | 131.2 ± 42.45 | 135.4 ± 49.37 | 136.47 ± 50.02 | 136.46 ± 51.52 | 135.8 ± 50.4 | 136.54 ± 51.03 | 138.07 ± 53.4 | <.0001 |
| Total Cholesterol, mg/dL | 187.84 ± 40.49 | 188.76 ± 40.6 | 189.1 ± 41.75 | 189.47 ± 42.22 | 191.52 ± 41.17 | 192.08 ± 41.24 | 191.84 ± 41.33 | <.0001 |
| HDL -C, mg/dL | 51.53 ± 14.9 | 49.68 ± 16.03 | 49.86 ± 16.2 | 49.29 ± 15.16 | 49.28 ± 14.52 | 49.48 ± 15.79 | 49.66 ± 16 | <.0001 |
| LDL -C, mg/dL | 107.1 ± 36.58 | 105.21 ± 37.96 | 105.64 ± 37.35 | 104.87 ± 38.61 | 106.13 ± 38.23 | 106.55 ± 38.69 | 105.96 ± 37.85 | <.0001 |
| eGFR, mL/min/1.73 m² | 85.22 ± 38.14 | 89.68 ± 41.92 | 88.84 ± 47.29 | 90.63 ± 43.87 | 92.1 ± 46.97 | 92.29 ± 48.01 | 92.65 ± 48.64 | <.0001 |
| **Smoking amount (1st Exam)** | | | | | | | | <.0001 |
| mild: < 10 cigarettes/d | – | 28456(50.25) | 13951(17.48) | 1335(4.01) | 2969(5.49) | 6774(3.67) | 14670(21.86) | |
| moderate: 10–19 cigarettes/d | – | 10695(18.88) | 30366(38.04) | 6642(19.93) | 17623(32.57) | 57690(31.26) | 36979(55.09) | |
| heavy: ≥ 20 cigarettes/d | – | 17482(30.87) | 35512(44.49) | 25350(76.06) | 33510(61.94) | 120101(65.07) | 15472(23.05) | |
| **Smoking duration (1st Exam)** | | | | | | | | <.0001 |
| <5 | – | 26530(46.85) | 3610(4.52) | 687(2.06) | 837(1.55) | 2713(1.47) | 2233(3.33) | |

*(Continued)*

| | | Never | Starter | Quitter | 50% Reducer | 20% Reducer | Sustainer | Increaser | P value |
|---|---|---|---|---|---|---|---|---|---|
| 5—9 | | – | 1633(2.88) | 3744(4.69) | 1182(3.55) | 1647(3.04) | 5114(2.77) | 3047(4.54) | |
| 10—19 | | – | 7326(12.94) | 15426(19.32) | 6237(18.71) | 11440(21.15) | 37236(20.18) | 16150(24.06) | |
| 20 | | – | 21144(37.34) | 57049(71.46) | 25221(75.68) | 40178(74.26) | 139502(75.58) | 45691(68.07) | |
| **Pack-years (1st Exam)** | | | | | | | | | <.0001 |
| <10 | | – | 32841(57.99) | 22050(27.62) | 4092(12.28) | 7039(13.01) | 25003(13.55) | 23117(34.44) | |
| 10 to <20 | | – | 7721(13.63) | 20415(25.57) | 6796(20.39) | 13296(24.58) | 47532(25.75) | 23017(34.29) | |
| 20 to <30 | | – | 6467(11.42) | 15474(19.38) | 7380(22.14) | 11839(21.88) | 49064(26.58) | 11201(16.69) | |
| ≥30 | | – | 9604(16.96) | 21890(27.42) | 15059(45.19) | 21928(40.53) | 62966(34.12) | 9786(14.58) | |
| **Smoking amount (2nd Exam)** | | | | | | | | | <.0001 |
| mild: < 10 cigarettes/d | | – | 11247(19.86) | 31853(39.9) | 11552(34.66) | 7562(13.98) | 6797(3.68) | 3956(5.89) | |
| moderate: 10–19 cigarettes/d | | – | 22035(38.91) | 18298(22.92) | 18042(54.14) | 30126(55.68) | 57748(31.29) | 19498(29.05) | |
| heavy: ≥ 20 cigarettes/d | | – | 23351(41.23) | 29678(37.18) | 3733(11.2) | 16414(30.34) | 120020(65.03) | 43667(65.06) | |
| **Smoking duration (2nd Exam)** | | | | | | | | | <.0001 |
| <5 | | – | 4371(7.72) | 28146(35.25) | 1137(3.41) | 759(1.4) | 1919(1.04) | 1203(1.79) | |
| 5-9 | | – | 2682(4.74) | 2106(2.64) | 1442(4.33) | 1375(2.54) | 3860(2.09) | 1997(2.98) | |
| 10-19 | | – | 10272(18.14) | 10412(13.04) | 6670(20.01) | 10715(19.81) | 31612(17.13) | 12879(19.19) | |
| ≥20 | | – | 39308(69.41) | 39165(49.06) | 24078(72.25) | 41253(76.25) | 147174(79.74) | 51042(76.04) | |
| **Pack-years (2nd Exam)** | | | | | | | | | <.0001 |
| <10 | | – | 17962(31.72) | 37743(47.28) | 14324(42.98) | 11515(21.28) | 21642(11.73) | 8902(13.26) | |
| 10 to <20 | | – | 14487(25.58) | 12661(15.86) | 11430(34.3) | 18717(34.6) | 44659(24.2) | 15566(23.19) | |
| 20 to <30 | | – | 10470(18.49) | 10732(13.44) | 4957(14.87) | 11549(21.35) | 48980(26.54) | 14521(21.63) | |
| ≥30 | | – | 13714(24.22) | 18693(23.42) | 2616(7.85) | 12321(22.77) | 69284(37.54) | 28132(41.91) | |

CKD: Chronic kidney disease, OHA: Oral hypoglycemic agents, BMI: Body mass index, SBP: Systolic blood pressure, DBP: Diastolic blood pressure.

## Association between smoking status, change in smoking intensity, and risk of HS

During the median IQR follow-up period of 5.2 (4.1–6.1) years, there were 2,102 HS events. Table 2 shows the hazard ratio (HR) of HS based on smoking status. Before adjustment, in the initial health checkup, the HR for HS among current smokers was 1.42 (95% CI, 1.29–1.57, unadjusted model). After adjusting for variables, the HR for current smokers was 1.23 (95% CI, 1.08–1.39, model 4), indicating an increased risk compared to nonsmokers and former smokers. Additionally, current smoking was associated with a 21% increased risk of HS during the 2-year follow-up examination (aHR, 1.21; 95% CI, 1.07–1.38, model 4). The results of the Cox regression analyses are presented in Table 3. A higher risk of HS was found in the sustainer (aHR, 1.236; 95% CI, 1.075–1.421) and increaser (aHR, 1.285; 95% CI, 1.048–1.577) groups compared with nonsmoking group. HRs of other groups including starter, quitter, and 50% reducer and 20% reducer were not significant.

Kaplan-Meier estimates showed that the incidence of HS was higher in increasers than in quitters. (Fig 3)

## Analysis stratified by BMI

There was no significant difference in the association between smoking intensity change and risk of HS categorized by BMI, even though no significant results were found in BMI ≥ 25 kg/m$^2$ due to the small number analyzed in the study (Table 4).

**Table 2. Risk of HS in DM according to smoking status.**

| Smoking status | N | (%) | HS event | Duration, Person-years | IR, 1000 Person-years | aHR (95% C.I) | | | | | | | |
|---|---|---|---|---|---|---|---|---|---|---|---|---|---|
| | | | | | | Model 1 | p-value | Model 2 | p-value | Model 3 | p-value | Model 4 | p-value |
| **Initial examination** | | | | | | | <.0001 | | 0.0051 | | 0.001 | | 0.0011 |
| Non | 957893 | 56.2 | 1051 | 4890535.31 | 0.2149 | 1 (Ref.) | | 1 (Ref.) | | 1 (Ref.) | | 1 (Ref.) | |
| Former | 328590 | 19.3 | 413 | 1645213.19 | 0.25103 | 1.16 (1.04, 1.31) | | 1.01 (0.88, 1.15) | | 1.01 (0.88, 1.16) | | 1.01 (0.88, 1.15) | |
| Current | 418944 | 24.6 | 638 | 2075405.92 | 0.30741 | 1.42 (1.29, 1.57) | | 1.19 (1.06, 1.35) | | 1.23 (1.08, 1.39) | | 1.23 (1.08, 1.39) | |
| **Second examination** | | | | | | | <.0001 | | 0.0277 | | 0.0056 | | 0.0057 |
| Non | 958192 | 56.2 | 1051 | 4882195.08 | 0.21527 | 1 (Ref.) | | 1 (Ref.) | | 1 (Ref.) | | 1 (Ref.) | |
| Former | 351487 | 20.6 | 455 | 1767536.25 | 0.25742 | 1.19 (1.07, 1.33) | | 1.03 (0.90, 1.17) | | 1.04 (0.91, 1.18) | | 1.03 (0.90, 1.18) | |
| Current | 395748 | 23.2 | 596 | 1961423.09 | 0.30386 | 1.41 (1.27, 1.55) | | 1.17 (1.03, 1.33) | | 1.21 (1.07, 1.38) | | 1.21 (1.07, 1.38) | |

Model 1: Not adjusted;

Model 2: Age and sex-adjusted;

Model 3: Model 2+Adjusted for income, drinking, regular exercise, hypertension, and dyslipidemia;

Model 4: Model 3+CKD, fasting glucose, DM duration, OHA use, insulin use

CKD: chronic kidney disease, OHA: oral hypoglycemic agents

**Table 3. Association between changes in smoking intensity and risk of HS.**

| | N | HS event | Duration, person-years | IR, 1000 Person-years | aHR (95% CI) | | | | | | | |
|---|---|---|---|---|---|---|---|---|---|---|---|---|
| | | | | | Model 1 | p-value | Model 2 | p-value | Model 3 | p-value | Model 4 | p-value |
| **Smoking change** | | | | | | <.0001 | | 0.0454 | | 0.0116 | | 0.0115 |
| Never smoker | 1229850 | 1398 | 6250268.57 | 0.22367 | 1 (Ref.) | NA | 1 (Ref.) | NA | 1 (Ref.) | NA | 1 (Ref.) | NA |
| Starter | 56633 | 66 | 285479.94 | 0.23119 | 1.032 (0.806, 1.321) | NA | 0.907 (0.706, 1.166) | NA | 0.932 (0.724, 1.198) | NA | 0.930 (0.724, 1.197) | NA |
| Quitter | 79829 | 108 | 399462.77 | 0.27036 | 1.207 (0.993, 1.468) | NA | 1.067 (0.873, 1.304) | NA | 1.077 (0.881, 1.316) | NA | 1.073 (0.878, 1.312) | NA |
| 50% reducer | 33327 | 53 | 163633.51 | 0.32389 | 1.443 (1.097, 1.898) | NA | 1.260 (0.954, 1.664) | NA | 1.295 (0.980, 1.711) | NA | 1.293 (0.979, 1.709) | NA |
| 20% reducer | 54102 | 84 | 268484.87 | 0.31287 | 1.394 (1.118, 1.737) | NA | 1.190 (0.949, 1.493) | NA | 1.227 (0.977, 1.540) | NA | 1.226 (0.976, 1.539) | NA |
| Sustainer | 184565 | 286 | 913593.98 | 0.31305 | 1.394 (1.227, 1.583) | NA | 1.193 (1.039, 1.369) | NA | 1.236 (1.075, 1.420) | NA | 1.236 (1.075, 1.421) | NA |
| Increaser | 67121 | 107 | 330230.8 | 0.32402 | 1.442 (1.185, 1.755) | NA | 1.237 (1.009, 1.516) | NA | 1.285 (1.047, 1.576) | NA | 1.285 (1.048, 1.577) | NA |

Model 1: Non-Adjusted.

Model 2: Age and sex-adjusted.

Model 3: Model 2+Income, drinking, regular exercise, hypertension, and dyslipidemia-adjusted.

Model 4: Model 3+CKD, fasting glucose, DM duration, OHA use, insulin use.

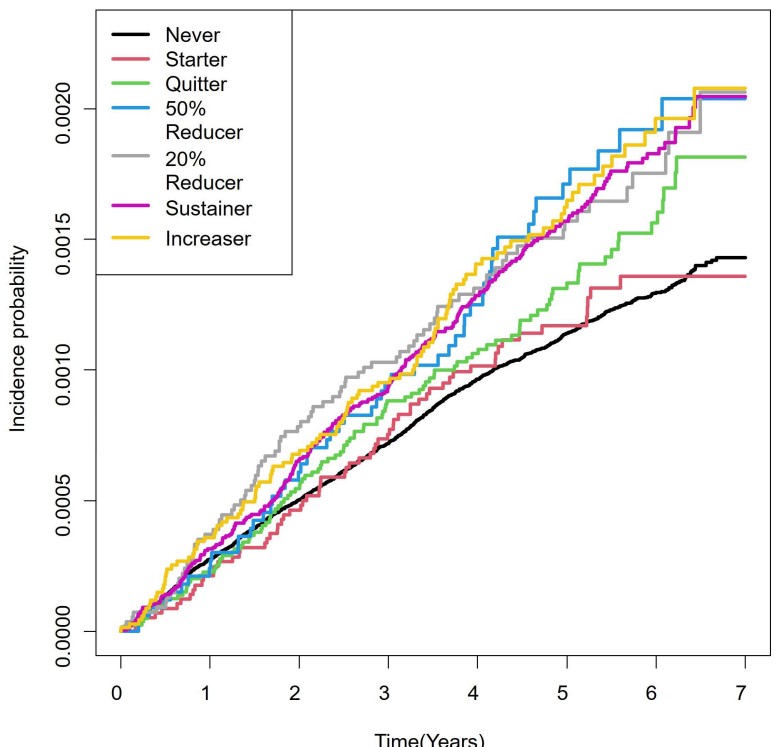

**Fig 3. Kaplan-Meier estimates for the probability of incident Hidradenitis suppurativa (HS) according to smoking cessation.**

**Table 4. Association between change in smoking intensity and risk of HS stratified by BMI.**

| | BMI < 25 kg/m2 | | | | | BMI ≥ 25 kg/m2 | | | | | *P* value |
|---|---|---|---|---|---|---|---|---|---|---|---|
| | N | HS event | Duration, person-years | IR, 1000 PY | aHR (95% CI) | N | HS event | Duration, person-years | IR, 1000 PY | aHR (95% CI) | |
| **Smoking change** | | | | | | | | | | | 0.0972 |
| Never smoker | 650327 | 748 | 3298567.87 | 0.22677 | 1 (Ref.) | 579523 | 650 | 2951700.7 | 0.22021 | 1 (Ref.) | NA |
| Starter | 31276 | 30 | 157188.25 | 0.19085 | 0.762 (0.528, 1.102) | 25357 | 36 | 128291.68 | 0.28061 | 1.138 (0.811, 1.598) | NA |
| Quitter | 42392 | 55 | 210993.62 | 0.26067 | 1.025 (0.777, 1.353) | 37437 | 53 | 188469.15 | 0.28121 | 1.129 (0.850, 1.499) | NA |
| 50% reducer | 19046 | 25 | 92859.43 | 0.26922 | 1.070 (0.716, 1.598) | 14281 | 28 | 70774.08 | 0.39563 | 1.584 (1.081, 2.320) | NA |
| 20% reducer | 29919 | 36 | 148341.87 | 0.24268 | 0.948 (0.675, 1.331) | 24183 | 48 | 120143 | 0.39952 | 1.570 (1.165, 2.118) | NA |
| Sustainer | 103441 | 155 | 512125.65 | 0.30266 | 1.188 (0.991, 1.426) | 81124 | 131 | 401468.32 | 0.3263 | 1.293 (1.062, 1.574) | NA |
| Increaser | 37125 | 50 | 182116.65 | 0.27455 | 1.085 (0.810, 1.453) | 29996 | 57 | 148114.15 | 0.38484 | 1.528 (1.158, 2.016) | NA |

## Discussion

A direct relationship between the severity of HS and smoking was previously presented [28]. For example, a study reported that nonsmokers showed an 11% higher rate of remission than current smokers [30]. Moreover, smoking cessation was associated with a lower rate of new lesion formation after radical excisional surgery in HS patients [46]. While such studies have explored smoking cessation as a modifiable risk factor for HS, to our knowledge, the link between

changes in smoking amounts and the predisposition to HS has not been examined in prior research. Therefore, this study investigated the association between changes in smoking levels and the risk of HS in patients with T2DM.

Among those who responded to this survey, 43% were active or former smokers. Notably, DM patients tend to have a lower smoking rate compared to the general population [47]. In the current study, smoking continuation was associated with a 23.6% increased risk of HS compared to never smokers over the two health examinations conducted at a 2-year interval. Furthermore, participants who increased their cigarette use showed a 28.5% increase in risk of HS. Two extensive retrospective studies in North America [24] and Israel [17], along with a single-center case-control study [13], also identified a significant association between smoking and HS using electronic records. Notably, one study reported a 90% increased risk (adjusted odds ratio, 1.9; 95% confidence interval, 1.8–2.0) of new HS diagnosis in smokers compared with nonsmokers, suggesting smoking as a potential risk factor for HS [24].

The relationship between smoking and HS remains an area of significant debate. While smoking is often suggested as a potential trigger for HS, its exact mechanism has yet to be clearly defined. One proposed pathway involves alterations in the chemotaxis of polymorphonuclear neutrophils, a mechanism similar to that in palmoplantar pustulosis [26,48]. Additionally, Kurzen et al.[49] proposed potential mechanisms for how cigarette smoking might contribute to the development of HS. Alkaloids in cigarette smoke may selectively inhibit most microorganisms while promoting the growth and proliferation of *Staphylococcus aureus*. This could potentially create a feedback loop altering the skin microbiome. Moreover, nicotine suppresses the production of antimicrobial peptides such as human beta-defensin 2, making follicles more vulnerable to bacterial invasion [50]. However, the role of *S. aureus* is complicated by its low prevalence in active HS lesions, suggesting that its involvement might occur in preclinical stages before symptoms manifest.

Another hypothesis involves the prolonged release of nicotine through sweat. Specifically, nicotine is emphasized as a modulator of the inflammatory response, contributing to various pathologies characterized by a strong immunological basis. These may include conditions like atherosclerosis, rheumatoid arthritis, inflammatory bowel disease, and skin disorders [2,51,52]. Nicotine interacts with nicotinic acetylcholine receptors (nAChRs) present on various cell types involved in hidradenitis suppurativa (HS) lesions, including keratinocytes, neutrophils, lymphocytes, macrophages, mast cells, and sebocytes [49,53]. In the epidermis of HS patients, nAChRs are strongly expressed around the pilosebaceous unit, contributing to excessive growth of the infundibular epithelium and the blockage of hair follicles [54]. Nicotine is also known to stimulate the release of pro-inflammatory cytokines like TNF-α [55] and inhibit the activity of immune cells such as macrophages and lymphocytes. This impairment may hinder the resolution of inflammation, contributing to the chronic nature of HS [49].

Additionally, cigarette smoke also contains polycyclic aromatic hydrocarbons (PAHs) and dioxin-like chemicals that activate the aryl hydrocarbon receptor (AhR) on keratinocytes, sebocytes, and immune cells [56–58]. In the context of HS, AhR activation may promote the expansion of Th17 cells [59], which are involved in chronic inflammatory conditions like Crohn's disease [60]. Also, PAHs can disrupt the normal differentiation of follicular keratinocytes, leading to the formation of comedones [61,62]. Overall, the combination of nicotine, PAHs and other chemicals from smoking can create a highly inflammatory environment, which may promote the pathogenesis of HS through both direct cellular effects and modulation of immune responses. The increased risk of HS in the sustainer and increaser group in this study supports these suggested mechanisms.

Meanwhile, there is a potential connection between hyperglycemia, insulin resistance, and diabetes, which may contribute to the development or worsening of HS [21]. Hyperglycemia is known to overactivate pathways associated with the mammalian target of rapamycin complex 1, a process implicated in keratinocyte proliferation, sebaceous gland proliferation, and lipogenesis [63,64]. This activation may predispose individuals to manifest clinical features of HS, such as follicular occlusion, sinus tract formation, and eventual scarring [65,66]. Although previous studies have refrained from drawing causal conclusions, they strongly support a notable and independent association between HS and diabetes [21].

Interestingly, no difference was observed in the association between changes in smoking intensity and the risk of HS according to BMI. Across seven studies, patients with HS were found to have a higher prevalence of obesity compared to control individuals, with prevalence rates ranging from 5.9% to 73.1% [13,17,67–71]. A meta-analysis involving eight studies reported that the adjusted pooled odds of obesity among HS patients were 3.5 times higher (95% CI, 2.2–5.4) than those of control individuals [72]. Elevated BMI is recognized as a potential risk factor for HS [71,73,74], and screening for obesity is recommended in HS patients [75]. Studies have also suggested a correlation between the severity of HS and BMI, with higher BMI levels being associated with more severe cases [28,76]. For instance, a cohort study in Italy found that obesity independently contributed to the severity of HS, as shown in both univariate and multivariate analyses [19]. Additionally, a survey suggested that individuals who quit smoking were more likely to achieve remission, particularly among those with normal weight, although a similar trend was observed in overweight patients as well [30]. In this study, no significant association was observed between changes in smoking intensity and the risk of HS across different BMI categories. This finding may be partly attributable to the limited number of HS events, which could have reduced the statistical power to detect meaningful differences and increased the likelihood of a Type II error. Furthermore, a relatively homogeneous or skewed BMI distribution among participants may have constrained the ability to observe differential effects across BMI strata. Beyond these methodological considerations, the interplay between smoking, BMI, and HS risk is inherently complex and may involve multifactorial biological mechanisms that are not yet fully understood. For instance, inflammation, hormonal regulation, and metabolic dysregulation—factors that are independently influenced by both smoking and adiposity—may confound or mediate the observed associations. These underlying processes could obscure direct relationships in stratified analyses. Consequently, future studies with larger sample sizes and more diverse populations are warranted to elucidate these interactions and clarify the potential modifying role of BMI in the association between smoking behavior and HS risk.

The present study indicates that maintaining cigarette smoking increases the risk of HS, while increasing the amount of smoking exacerbates this risk. The impact of smoking reduction on health outcomes may vary depending on the underlying causes. For example, smoking reduction has been linked to a decreased risk of certain cancers, such as lung cancer, but not necessarily to a lower risk of cardiovascular diseases [38,77].

Several limitations of the study should be noted. First, smoking data were self-reported, introducing potential subjectivity as individuals might underestimate their cigarette consumption. Second, since the recognition of HS was derived from nationwide claims data, access to clinical details and imaging results was limited. This raises the possibility of misdiagnosing abscesses as HS. However, given the recurrent nature of HS, the impact of misdiagnosis is likely minimal. Third, the observational nature of the study precludes establishing causal relationships between the variables. Additionally, the possibility of unmeasured confounders cannot be excluded. Lastly, it is essential to note that this study exclusively involved Korean participants, emphasizing the necessity for validation in diverse ethnic populations.

The findings suggest that reducing smoking does not directly lower the risk of HS. This lack of association may stem from the multifactorial pathogenesis of HS, suggesting it does not exhibit a dose-response relationship with cigarette consumption. It could also be due to the relatively short follow-up period. Nevertheless, the absence of an increased risk of HS in reducers and quitters underscores the importance of smoking reduction as a crucial initial step toward complete cessation.

## Author contributions

**Conceptualization:** Kyungdo Han.

**Data curation:** Gyu-Na Lee.

**Formal analysis:** Gyu-Na Lee.

**Supervision:** Ji Hyun Lee.

**Validation:** Esther Kim.

**Writing – original draft:** Hwa Jung Yook.

**Writing – review & editing:** Yeong Ho Kim, Ji Hyun Lee.

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
