## [Decision Letter · Decision Letter 0]

30 Oct 2024

PONE-D-24-40744Hidradenitis suppurativa (HS) in diabetes mellitus patients: A link between alterations in smoking intensity and the risk of developing HSPLOS ONE

Dear Dr. Lee,

Thank you for submitting your manuscript to PLOS ONE. After careful consideration, we feel that it has merit but does not fully meet PLOS ONE’s publication criteria as it currently stands. Therefore, we invite you to submit a revised version of the manuscript that addresses the points raised during the review process.

We look forward to receiving your revised manuscript.

Kind regards,

Yogesh Chander, Ph.D.

Academic Editor

PLOS ONE

**Journal Requirements:**

2. In the online submission form, you indicated that The data in this study was obtained from KNHIS where license may apply. Further enquiries can be directed to the corresponding author.

Reviewers' comments:

Reviewer's Responses to Questions

**Comments to the Author**

1. Is the manuscript technically sound, and do the data support the conclusions?

Reviewer #1: Yes

Reviewer #2: Partly

Reviewer #3: Yes

Reviewer #4: Yes

2. Has the statistical analysis been performed appropriately and rigorously? 

Reviewer #1: I Don't Know

Reviewer #2: Yes

Reviewer #3: Yes

Reviewer #4: Yes

3. Have the authors made all data underlying the findings in their manuscript fully available?

Reviewer #1: No

Reviewer #2: Yes

Reviewer #3: Yes

Reviewer #4: Yes

4. Is the manuscript presented in an intelligible fashion and written in standard English?

Reviewer #1: Yes

Reviewer #2: Yes

Reviewer #3: Yes

Reviewer #4: Yes

5. Review Comments to the Author

**Reviewer #1:**  Thank you for inviting me to review this paper. The authors have investigated the association of alterations in smoking intensity and the risk of developing HS in patients with DM. Overall, the association of smoking and HS has been previously studied. Thus, the topic is not a novel one. However, the finding regarding BMI is interesting. The method section needs to be improved as it requires more detail.

My comments are the followings:

-line 39: compared to which population?

-line 40: please clarify that you are discussing smoking.

-line 47: which year?

-did the authors exclude patients using anti-diabetic medications for reasons other than DM?

-add more details on blood sample collection and measurement methods.

-add more details on height and weight measurement for BMI calculation.

-cite the reference associated with physical activity categories.

- line 165: which figure? also add numerical values.

- mention the observational nature of the study as limitation for finding causation.

-define abbreviations below the tables.

-what is the possible explanation for finding no association after BMI stratification?

**Reviewer #2: ** It is suggested that the manuscript should replace instances of "we" or "our" with phrases such as "current study," "this study," or "present study" to minimize repetition.

The rationale of the study should be clearly stated at the end of the introduction.

Extensive proofreading and rephrasing are essential.

The discussion should be more concise and include more references for better comparison with previous research.

Substantial paraphrasing and thorough proofreading are necessary.

The manuscript has low quality. It did not address any specific gap they intended to fill. Furthermore, it does not add anything to the subject area compared with other published Material. Therefore, any comments regarding the methodology are equivocal. This manuscript does not fulfill the standards established for the journal to be considered for publication.

**Reviewer #3: ** 1. Conclusion: "This study found an association between smoking and the risk of HS, with implications for targeted interventions and smoking cessation strategies in at-risk populations". The conclusion should be less related to the target group of the study (diabetics).

Results:

2. Please provide a flow diagram and report numbers of individuals at each stage of study—eg numbers potentially eligible and give reasons for non-participation at each stage.

3. Can medication for diabetes affect the association of Smoking on HS? Is this not examined in this study?

4. The median/mean follow-up time should be mentioned

5. In the title of the first table, you must also specify the smoking status as in the second table.

6. In Table 3-4, the p value should be reported according to the number of risk ratio estimates (n=6 estimations) in last column.

7. In table 4, you can also do the trend test

**Reviewer #4:**  Suggested title- A link between smoking behaviours and the risk of Hidradenitis Suppurative in diabetic patients

Abstract-is too simplified and does not explore why the team wanted to study HS in DM and link with smoking.

Intro-intro talks more about HS-DM cause-effect relationship and touched only abit about smoking & HS

- line 63: Two meta-analyses found a 1.69–3.00 fold increase in the odds of DM in HS populations [10, 23, 24]. (added fold)

- line 68: Therefore, we aimed to investigate the effects of changes in smoking on the risk of HS in patients with DM using a large-scale, nationally-representative claims database in Korea. (fragmented sentence, suggest revise)

- the link between DM-Smoking-HS remains elusive and why this is the focus of the study is still unclear from the literature reviews done.

Materials & methods:

- need proofreading by a native speaker who are familiar with health related research manuscript.

- some confounding factors not explored: anti-inflammatory medication, surgical procedures, other medications/ treatment for HS, and diabetic control (HBA1c levels) need to be mentioned.

- Definition of smoking intensity change :suggest to change to definition of a change in smoking intensity

- Definition of never, former and current smokers were unclear, reducers, sustainers, increasers and starters? These definition may be better presented in a table. Would be good to get this presented as part of an experimental design flowchart (as a figure)

line 126: All blood samples were taken during fasting

Results

- line: 145 The cohort comprised of 1,705,427 participants (581,618 males [34.1%], 1,123,809 females [65.9%]; mean [SD] age, 59.03 [11.76] years). (added of)

- line 165: Kaplan-Meier estimates showed that the incidence of HS was higher in increasers than in quitters. (Figure) (which figure? Each figure should have a number assigned)

Discussion

- Minor proofreading help with grammar such as putting connectors where appropriate

- Would be good to discuss the effects of polyaromatic carbohydrates and nicotine in HS development to better link tobacco smoking and HS

- Would like to suggest the authors to refrain from using first-person language in the discussion to keep the discussion more impersonal.

- Limitation: HS treatment can be many, and over the course of 2 years, participants may have taken multiple medications to prevent or treat the disease which may lead to bias in the findings. Medication data was not available and should be mentioned in the limitation.

Tables and figures:

- suggest to refer to "figure" as "figure 1"

- suggest to increase figure 1 image quality to 300 dpi

- suggest to include experimental design flowchart in the form of a figure

6. PLOS authors have the option to publish the peer review history of their article (what does this mean? ). If published, this will include your full peer review and any attached files.

**Do you want your identity to be public for this peer review?** For information about this choice, including consent withdrawal, please see our Privacy Policy .

Reviewer #1: No

Reviewer #2: **Yes: ** Shooka Mohammadi

Reviewer #3: No

Reviewer #4: **Yes: ** Intan Suhana Zulkafli

---

## [Author Response · Author response to Decision Letter 1]

12 Dec 2024

Thank you very much for your kind editorial letter.

We have attempted to carefully and thoroughly address all concerns raised by the editors and reviewers. With the help of your suggestions, we believe our manuscript has significantly improved.

We trust that we therewith have fulfilled all the Editor’s and Reviewer’s requests.

Thank you very much for your consideration.

---

## [Decision Letter · Decision Letter 1]

4 Apr 2025

PONE-D-24-40744R1A link between smoking behaviors and the risk of Hidradenitis Suppurativa in diabetic patientsPLOS ONE

Dear Dr. Lee,

Thank you for submitting your manuscript to PLOS ONE. After careful consideration, we feel that it has merit but does not fully meet PLOS ONE’s publication criteria as it currently stands. Therefore, we invite you to submit a revised version of the manuscript that addresses the points raised during the review process.

The revised manuscript is re-evaluated by the previous reviewers. Please address reviewers' remaining concerns.

We look forward to receiving your revised manuscript.

Kind regards,

Jianhong Zhou

Staff Editor

PLOS ONE

Journal Requirements:

Reviewers' comments:

Reviewer's Responses to Questions

**Comments to the Author**

1. If the authors have adequately addressed your comments raised in a previous round of review and you feel that this manuscript is now acceptable for publication, you may indicate that here to bypass the “Comments to the Author” section, enter your conflict of interest statement in the “Confidential to Editor” section, and submit your "Accept" recommendation.

Reviewer #1: (No Response)

Reviewer #2: (No Response)

Reviewer #3: All comments have been addressed

Reviewer #4: All comments have been addressed

2. Is the manuscript technically sound, and do the data support the conclusions?

Reviewer #1: (No Response)

Reviewer #2: No

Reviewer #3: Yes

Reviewer #4: Yes

3. Has the statistical analysis been performed appropriately and rigorously? 

Reviewer #1: (No Response)

Reviewer #2: No

Reviewer #3: Yes

Reviewer #4: Yes

4. Have the authors made all data underlying the findings in their manuscript fully available?

Reviewer #1: (No Response)

Reviewer #2: Yes

Reviewer #3: Yes

Reviewer #4: Yes

5. Is the manuscript presented in an intelligible fashion and written in standard English?

Reviewer #1: (No Response)

Reviewer #2: Yes

Reviewer #3: Yes

Reviewer #4: Yes

6. Review Comments to the Author

Reviewer #1: -Add reference for the Korean Association of Laboratory Quality Control guidelines.

-Please add the possible explanation for finding no association after BMI stratification to the discussion section as well.

Reviewer #2: The manuscript has low quality. It failed to meet the established standards required for consideration in the journal.

Reviewer #3: Dear editor,

All comment have been addressed by authors, My Final Decision is ACCEPTANCE.

Best Regards,

Cheraghi

Reviewer #4: The authors have addressed all the comments from all reviewers satisfactorily. The authors responded to each individual suggestions with good explanation supplemented by figures. These have improved the manuscript significantly.

7. PLOS authors have the option to publish the peer review history of their article (what does this mean? ). If published, this will include your full peer review and any attached files.

**Do you want your identity to be public for this peer review?** For information about this choice, including consent withdrawal, please see our Privacy Policy .

Reviewer #1: No

Reviewer #2: No

Reviewer #3: No

Reviewer #4: **Yes: ** INTAN SUHANA ZULKAFLI

---

## [Author Response · Author response to Decision Letter 2]

28 Apr 2025

To

Editor of PLOS ONE

Submission ID d914ed04d190a2b4

"Hidradenitis suppurativa (HS) in diabetes mellitus patients: A link between alterations in smoking intensity and the risk of developing HS"

Thank you very much for your kind editorial letter.

We have attempted to carefully and thoroughly address all concerns raised by the editors and reviewers. With the help of your suggestions, we believe our manuscript has significantly improved.

We trust that we therewith have fulfilled all the Editor’s and Reviewer’s requests.

Thank you very much for your consideration.

---

## [Decision Letter · Decision Letter 2]

13 May 2025

A link between smoking behaviors and the risk of Hidradenitis Suppurativa in diabetic patients

PONE-D-24-40744R2

Dear Dr. Lee,

We’re pleased to inform you that your manuscript has been judged scientifically suitable for publication and will be formally accepted for publication once it meets all outstanding technical requirements.

Kind regards,

Emanuele Scala

Academic Editor

PLOS ONE

Additional Editor Comments (optional):

Reviewers' comments:

Reviewer's Responses to Questions

**Comments to the Author**

1. If the authors have adequately addressed your comments raised in a previous round of review and you feel that this manuscript is now acceptable for publication, you may indicate that here to bypass the “Comments to the Author” section, enter your conflict of interest statement in the “Confidential to Editor” section, and submit your "Accept" recommendation.

Reviewer #1: (No Response)

Reviewer #2: All comments have been addressed

Reviewer #4: All comments have been addressed

2. Is the manuscript technically sound, and do the data support the conclusions?

Reviewer #1: (No Response)

Reviewer #2: No

Reviewer #4: Yes

3. Has the statistical analysis been performed appropriately and rigorously? 

Reviewer #1: (No Response)

Reviewer #2: No

Reviewer #4: Yes

4. Have the authors made all data underlying the findings in their manuscript fully available?

Reviewer #1: (No Response)

Reviewer #2: Yes

Reviewer #4: Yes

5. Is the manuscript presented in an intelligible fashion and written in standard English?

Reviewer #1: (No Response)

Reviewer #2: Yes

Reviewer #4: Yes

6. Review Comments to the Author

Reviewer #1: (No Response)

Reviewer #2: The manuscript has low quality. It failed to meet the established standards required for consideration in the journal.

Reviewer #4: Abstract: massive improvement, well written with clear direction

Introduction: very well-written and edited

Methods:

overall the methodologies section is well written & edited with sound methodologies use

Line 109 is unclear/ hanging: Anti-diabetic medications included insulin, metformin, sulfonylureas, meglitinides, dipeptidyl peptidase 4 (DPP-4) inhibitors, thiazolidinediones, and α-glucosidase inhibitors.

Suggest change to:

Anti-diabetic medications included were insulin, metformin, sulfonylureas, meglitinides, dipeptidyl peptidase 4 (DPP-4) inhibitors, thiazolidinediones, and α-glucosidase inhibitors.

Results:

adequately written

Discussion:

comprehensively discussed

tables and images are in good quality

7. PLOS authors have the option to publish the peer review history of their article (what does this mean? ). If published, this will include your full peer review and any attached files.

**Do you want your identity to be public for this peer review?** For information about this choice, including consent withdrawal, please see our Privacy Policy .

Reviewer #1: No

Reviewer #2: No

Reviewer #4: **Yes: ** INTAN SUHANA ZULKAFLI

---

## [Editor Report · Acceptance letter]

PONE-D-24-40744R2

PLOS ONE

Dear Dr. Lee,

I'm pleased to inform you that your manuscript has been deemed suitable for publication in PLOS ONE. Congratulations! Your manuscript is now being handed over to our production team.

Kind regards,

on behalf of

Dr. Emanuele Scala

Academic Editor

PLOS ONE